# High-Sensitivity Pressure Sensors Based on a Low Elastic Modulus Adhesive

**DOI:** 10.3390/s22093425

**Published:** 2022-04-30

**Authors:** Xiuzhu Xu, Hao Zhu, Shengping Dai, Tao Sun, Guanggui Cheng, Jianning Ding

**Affiliations:** School of Mechanical Engineering, Institute of Intelligent Flexible Mechatronics, Jiangsu University, Zhenjiang 212013, China; xuxz20181231@163.com (X.X.); zh3412534@163.com (H.Z.); dsp@ujs.edu.cn (S.D.); suntujs@163.com (T.S.); ggcheng@ujs.edu.cn (G.C.)

**Keywords:** adhesives, high sensitivity, low elastic modulus, pressure sensors

## Abstract

With the rapid development of intelligent applications, the demand for high-sensitivity pressure sensor is increasing. However, the simple and efficient preparation of an industrial high-sensitivity sensor is still a challenge. In this study, adhesives with different elastic moduli are used to bond pressure-sensitive elements of double-sided sensitive grids to prepare a highly sensitive and fatigue-resistant pressure sensor. It was observed that the low elastic modulus adhesive effectively produced tensile and compressive strains on both sides of the sensitive grids to induce greater strain transfer efficiency in the pressure sensor, thus improving its sensitivity. The sensitivity of the sensor was simulated by finite element analysis to verify that the low elastic modulus adhesive could enhance the sensitivity of the sensor up to 12%. The preparation of high-precision and fatigue-resistant pressure sensors based on low elastic modulus, double-sided sensitive grids makes their application more flexible and convenient, which is urgently needed in the miniaturization and integration electronics field.

## 1. Introduction

Flexible multifunctional sensors represent a significant branch of the next generation of flexible electronics, exhibiting great potential for application in smart wear, human–computer interaction, and soft robotics [1,2,3,4,5,6,7,8]. Numerous types of pressure sensors already exist, such as capacitive inductive [9,10], piezoelectric ceramic [11,12], and resistive strain gauges [13,14]. These types of pressure sensors usually require complex circuit and mechanical structural designs [15]. The capacitive sensors require high accuracy in processing and assembly to achieve a certain degree of sensing accuracy, and they have poor anti-interference ability and parasitic capacitance problems. In contrast, piezoelectric ceramic pressure sensors require special installation methods, and they are susceptible to environmental influences, which significantly increase the operating cost and pose difficulties for large-scale production. Resistive strain sensors employ the resistive strain effect caused by changes in the conductive path generated by external forces on the sensitive unit, resulting in advantages such as simple production processes, strong anti-interference capability, and compatibility with a wide range of applications.

In order to further increase the sensitivity of the sensor, a micro-nano structure, different materials, and other methods have been used [16,17,18,19,20]. Lee et al. used single-walled carbon nanotube (SWCNT) strain gauges and epoxy adhesives to develop sensors that were able to achieve a high strain transfer efficiency owing to the high sensitivity coefficient (Ks = 59.2) of the gaps in the SWCNT arrangement; however, the fabrication process was very complex [21,22]. Few studies have applied low elastic modulus adhesives to strain gauges for strain measurement, as most people believe that low elastic modulus adhesives in a soft state have weaknesses, such as unfavorable strain transfer and poor adhesion [23,24,25,26,27,28,29]. In contrast to the conventional view, in this study, we observed that low-modulus adhesives are more effective in creating adequate tensile and compressive strains with the Wheatstone full bridge output than high-modulus adhesives in double-sided sensitive grid structure pressure sensors, thus achieving higher sensitivity.

The purpose of this study is to investigate the effect of different elastic modulus adhesives on the sensitivity coefficient of strain gauges and to understand their mechanism of action. In this study, the graphene conductive ink was used to print two sensitive grids on both sides of a polyimide (PI) film to create a pressure sensor based on the screen-printing process, which can effectively capture the small deformation of the glass screen and achieve touch-pressure sensing. This study focused on a sensor with double-sided sensitive grids and different pressure transmission elements made from 3M double-sided adhesive and Loctite 401. The sensitivity coefficients of the sensor with the double-sided sensitive grids and bonding with high and low elastic modulus adhesives were compared and analyzed through finite element simulations and experiments.

## 2. Materials and Methods

Experimental materials: The materials and models used in the experiments are shown in Table 1 below.

Experimental equipment: The equipment and models used in the experiments are shown in Table 2 below.

Manufacturing process: The screen-printed sensor developed using the abovementioned processes is shown in Figure 1. Firstly, the PI film with a suitable size was put on a vacuum table of the screen-printing machine, and a charge coupled device (CCD) camera was used to catch the position of the target on the screen-printing plate. Secondly, the conductive silver paste on the screen plate was printed on the PI film by scraper walking and extrusion. The conductive silver paste line on the screen was printed on the PI film. Next, the PI film with the newly printed conductive silver paste was put into in a vacuum oven and cured at 130 °C for about 30 min. Thirdly, the graphene-sensitive grids were printed in the same way. The PI film with the cured conductive silver paste was put on the vacuum table, and a CCD camera was used to catch the position of the target on the screen-printing plate in the sensitive grids’ pattern. The graphene conductive ink on the plate was printed on the PI film by scraper walking and extrusion. The two sensitive grid lines on the screen were printed on the PI film. Next, the PI film with the newly printed graphene conductive ink was put into in a vacuum oven and cured at cured at 150 °C for about 180 min. Fourthly, the UV adhesive was applied to the screen-printing plate in a covering layer pattern. The UV adhesive covered the graphene sensitive grids and most of the conductive silver paste line. Then PI film was put under a UV light to cure. Then, on the other side of the conductive silver line, graphene-sensitive grids and a UV adhesive protective layer were printed in the same order and operation.

Testing process: Firstly, a cross line should be drawn in the middle of the glass screen, and the center of the two sensitive grids of the sensor was aligned with the center of the cross line, and then the sensor was pasted into the middle surface of the glass screen with 3M glue. Secondly, the glass screen was placed on the mobile phone frame printed with a 3D printer, and the surface of the glass screen was faced up, whose surface was not pasted with the sensor. Thirdly, the loading-support fixture was placed on the glass screen at the center of the sensor, and then the wires on the sensor were lead to connect to the regulated power supply and Keithley resistance equipment meter. Finally, a 100 g weight was put on the loading support fixture, afterwards recording the value change of the Keithley meter.

## 3. Results and Discussion

In this study, graphene conductive ink was used as the conductive filler to form a resistive pressure sensor on the surface of the PI film by screen printing, as shown in Figure 2a–c. The surface structure of the resistive sensor was recorded by scanning electron microscopy (SEM), as shown in Figure 3a. We found that the surface of the film is smooth without obvious agglomeration, indicating that the graphene is uniformly dispersed in the conductive ink. The sensor measured resistance performance by connecting a regulated power supply and Keithley resistance equipment meter, with a value of approximately 463.32 Ω and where the standard variance was about 21.86 Ω. Furthermore, the resistance values of the screen-printed strain gauges were statistically analyzed with Minitab software [30]. The distribution of the resistance values of the screen-printed strain gauges, which are sensitive elements for gluing, were found to be normally distributed (*p* < 0.05), as shown in Figure 3b. The excellent resistance characteristics of the resistance sensor provide a basis for the subsequent study of the influence of adhesives on sensor sensitivity. There is a better consistence of the resistance, which provides the good suitability of the sensors compared to another literature study when it comes to mass production and cost-efficiency in the future [19].

To study the influence of adhesives with different modules of elasticity on the sensitivity output of the pressure sensors, two typical adhesives with relatively significant differences in their elastic moduli were selected for comparison: a 3M double-sided adhesive with a 0.4 MPa elastic modulus, and a Henkel Loctite 401 instant adhesive with approximately a 2.5 GPa elastic modulus (Appendix A). The sensor was attached to the back of the phone glass screen under 3 V regulated power supply and 100 g load [31] (Figure 2a,b). The mean full-bridge voltage output of the sensor (L4) was 0.000257 V when attached to the Loctite 401 adhesive, resulting in a sensitivity of 0.0856 mV/V. Meanwhile, the mean full-bridge voltage output of the sensor (M3) was found to be 0.000303 V when attached to the 3M double-sided tape, resulting in a sensitivity of 0.101 mV/V, as deduced from Appendix A. The full-bridge voltage output of M3 is approximately 1.179 times higher than the full-bridge voltage output of L4. So M3 is 17.9% more sensitive than L4 under the same pressure. In addition, the M3 sensor was tested for fatigue using a custom-made cam fatigue test rig, as shown in Figure 3a. The relative resistance change rate was observed to drift by less than 1% after one million continuous cycles under a 100 g load and a frequency of 1 Hz. This indicates that the sensor using 3M adhesive exhibits excellent stability performance.

To further understand the effect of the two adhesives on the sensitivity of the sensor, the above mobile phone press model is shown in Appendix A. The phone press model was constructed and analyzed by finite element simulation [32]. The results were illustrated in Figure 4b. When the 3M double-sided adhesive was applied to the sensor, the sensitive grids on the non-adhesive side of the sensor sensed tensile strain, and the other sides of the sensor sensed compressive strain. Whereas, the sensitive grid on both sides of the sensor sensed tensile strain when Loctite 401 adhesive was applied to the sensor. The tensile strain induced by the Loctite 401-bonded sensitive grids is greater than the absolute value of the strain induced by the 3M double-sided adhesive. It indicated the greater strain transfer efficiency obtained by the sensor with a larger elastic modulus. The difference between the tensile strain caused by the 3M double-sided adhesive acting on the sensitive grid and the compressive strain is larger than that induced by the Loctite 401 adhesive (Figure 4b). The large difference represents that the output of the strain signal is larger when forming the Wheatstone full bridge circuit. Through the calculation of the simulation results, the results of the finite element simulations demonstrated that the total strain output of the full bridge was 12% higher with M3 than with L4 under the same pressure (100 g), which is in line with the results of the experimental tests, which are shown in Appendix A. The simulation results and the calculation formula of sensitivity (Appendix A) are summarized in Table 3. The comparative total strain output and sensitivity of M3 and L4 are shown in the simulation and experimental test, respectively. Sensitive grids on both sides of the low elastic modulus adhesive were observed to effectively induce tensile and compressive strains for higher sensitivity. The low elastic modulus adhesive effectively improves the sensitivity of the pressure sensor with the double-sided sensitive grids, which will further broaden the structure, process, and application range of the pressure sensor. According to the total strain and sensitivity, the sensitivity factor K can be approximately calculated (Appendix A). The sensitivity factor K of M3 is 9.2 and the sensitivity factor K of L4 is 8.7. The sensitivity factor K of this study is better than the sandwich-structured strain sensor (GF = 1.5) [33], and it is similar to the result of the literature (GF = 7.7~9.4) [34]. The sensor also has a better response time (Appendix A) than Han et al., listed in Table 1 [19]. Furthermore, the sensors in this study are very suitable for mass production in the future and are cost efficient.

In order to obtain the influence rule of different elastic modulus adhesives on the total strain output, the simulation of adhesive-backed sensors with different elastic moduli was carried out. The total strain output is the difference between the strain on the two sensitive grids on the adhesive-backed surface and the strain on the two sensitive grids on the adhesive surface, and the calculation formula is presented in the Appendix A. As shown in Figure 5a, when the elastic modulus of the adhesive is less than 1 MPa, the total strain increases with the elastic modulus of the adhesive increases. The tensile strain on the backing surface is a positive strain, which increases as the elastic modulus of the adhesive increases. Whereas, the compressive strain on the adhesive surface is a negative strain, which decreases as the elastic modulus of the adhesive increases. When the elastic modulus of the adhesive ranged 1–500 MPa, the total strain decreased as the elastic modulus of the adhesive increased. The strain on both the backing surface (Non-Glue Surface) and the adhesive surface (Glue Surface) increased with the elastic modulus of the adhesive increasing, and the slope of the increase on the adhesive surface was greater than the slope of the increase on the backing surface (Figure 5b). When the elastic modulus of the adhesive was greater than 500 MPa, the total strain, and the strain on the backing and adhesive surfaces, were both positive strains, and they all tended to be constant.

Deformation occurs under the action of loading, and the soft adhesive (low elastic modulus adhesive) is connected tightly with the glass screen, and the soft adhesive close to the glass screen extends around the deformation of the glass screen, whereas the other side of the soft adhesive is connected tightly with the sensor, and the sensor hinders the deformation of the soft adhesive. Since the soft adhesive is a flexible body, the soft adhesive close to the glass screen extends around, resulting in the tightening of the middle part of the soft adhesive (necking phenomenon), and the contraction of the surface between the soft glue and sensor, which forms a contracted compressive strain on the sensor surface; that is, a compressive strain is formed on the sensor surface close to the soft glue. Meanwhile, the sensor bending induces a tensile strain formed on the other side of the sensor (Figure 5c). When the glass screen deforms under the action of the loading, the hard adhesive (high elastic modulus adhesive) connected tightly to the glass screen extends around the deformation of the glass screen. Because the hard adhesive layer is equivalent to a rigid body, the expansion of the hard adhesive close to the glass screen also exhibits a tensile effect on the sensor surface, which results in a tensile strain on the sensor surface, and the bending deformation of the sensor leads to a tensile strain on the other side of the sensor (Figure 5d). At the area around the hard adhesive occurs an incline phenomenon. Therefore, the strain difference of the adhesive layers with different elastic modulus under load has a serious impact on the sensitivity of the sensor.

## 4. Conclusions

In this paper, through the experimental method and finite element simulation method, the sensitivity coefficients of the sensor with double-sided sensitive grids adhered by the high and low elastic modulus adhesives are compared and analyzed. It is different from the traditional sensor with double-sided sensitive grids adhered by a high elastic modulus adhesive, as the surface in contact with the adhesive and the surface not in contact with the adhesive are subjected to the tensile strain synchronously. The sensor with a double-sided sensitive grid pasted with a low elastic modulus adhesive can form tensile and compressive strains on the two surfaces of the sensor, respectively, so as to achieve higher sensitivity (12%). The sensors in this study are also very suitable for mass production in the future and are cost efficient. We expect this convenient and efficient preparation method of a high-sensitivity sensor to greatly widen the sensor applications in industrial manufacturing and robots that require high sensitivity, thereby making a significant contribution to the development of human industry.

## Figures and Tables

**Figure 1 sensors-22-03425-f001:**
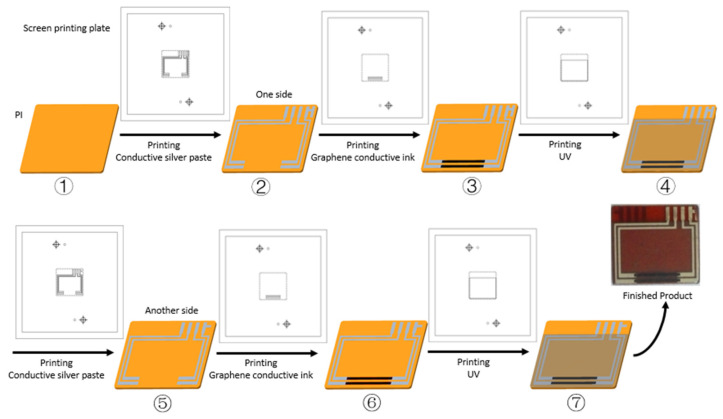
Schematic diagram of the manufacturing process of the sensor, ① PI film; ② Conductive silver paste line on the PI film; ③ Graphene-sensitive grids printed between conductive silver paste line; ④ UV adhesive covered the graphene sensitive grids and most of the conductive silver paste line; ⑤ In another side, conductive silver paste line on the PI film; ⑥ In another side, graphene-sensitive grids printed between conductive silver paste line; ⑦ In another side, UV adhesive covered the graphene sensitive grids and most of the conductive silver paste line.

**Figure 2 sensors-22-03425-f002:**
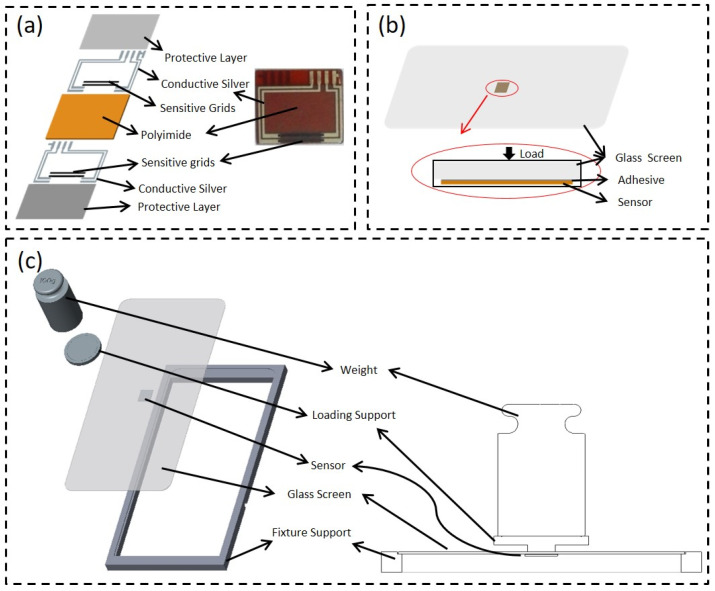
(**a**) Structure of the sensor; (**b**) schematic diagram of the arrangement of the adhesive and sensor on the glass screen; (**c**) schematic diagram of the loading status.

**Figure 3 sensors-22-03425-f003:**
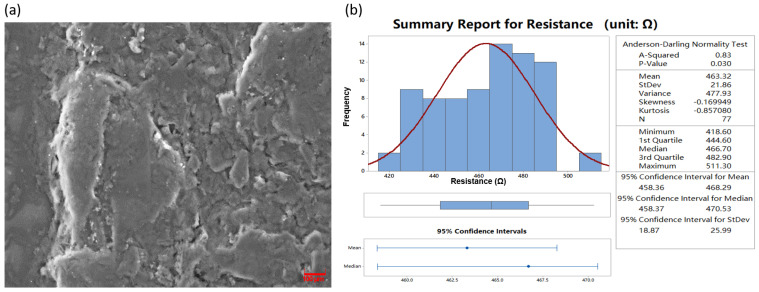
(**a**) The SEM picture of the resistive sensor; (**b**) distribution of the resistance values made by the screen printing.

**Figure 4 sensors-22-03425-f004:**
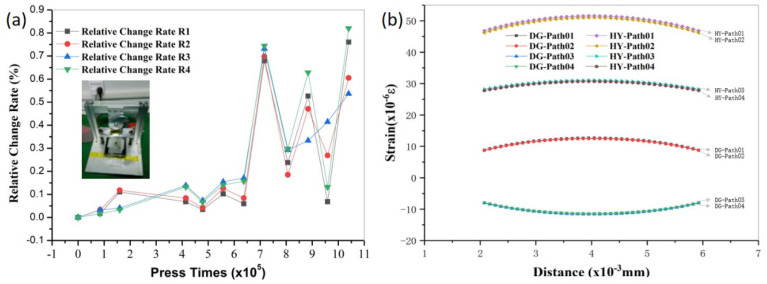
(**a**) Sensor resistance drift during the fatigue test; (**b**) strain distribution curves for the M3 and L4 sensitive grids.

**Figure 5 sensors-22-03425-f005:**
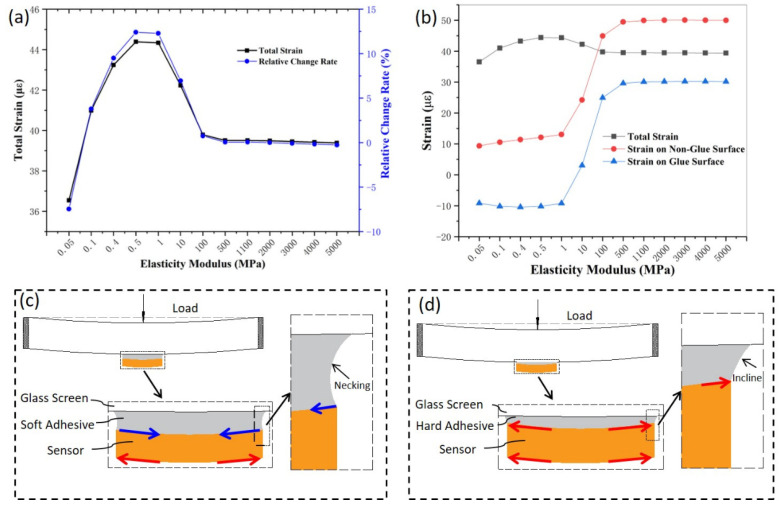
(**a**) Total strain output and relative rate of change versus the elastic modulus of the adhesive; (**b**) total strain, strain on the Non-Glue (non-adhesive) surface, and strain on the Glue (adhesive) surface versus the elastic modulus of the adhesive; (**c**) schematic of the force deformation of the low elastic modulus adhesive model; (**d**) schematic of the force deformation of the high elastic modulus adhesive model.

**Table 1 sensors-22-03425-t001:** Materials and models, and the companies that made them, used in the experiments.

Material Name	Model Name	Company
Graphene conductive ink	TJ02	Shanghai Enwang Material Technology Co., Ltd. (Shanghai, China)
Conductive silver paste	NT-ST80b	Beijing konaton Electronic Technology Co., Ltd. (Beijing, China)
Epoxy conductive adhesive	YC-01	Nanjing Xiliter Adhesive Co., Ltd. (Nanjing, China)
UV	KSM-180 g GH1	Jiangsu Guangxin New Photosensitive Materials Co., Ltd. (Jiangyin, China)
PI film	0.25 mm	Jiangsu Yabao Insulating Materials Co., Ltd. (Yangzhou, China)
Mobile phone glass screen	152.3 × 72.5 × 0.7 mm	Suzhou Grens Photoelectric Technology Co., Ltd. (Suzhou, China)
Screen printing plate	Polyester Screen	Changzhou Pratt Printing Technology Co., Ltd. (Changzhou, China)

**Table 2 sensors-22-03425-t002:** Equipment and models, and the companies that made them, used in the experiments.

Equipment Name	Model Name	Company
Screen-printing machine	PHP-2525	Shanghai Xuanting Screen-printing Equipment Co., Ltd. (Shanghai, China)
Vacuum drying oven	DZF-6020	Shanghai Bosun Industrial Co., Ltd. (Shanghai, China)
UV curing lamp	BHL-1000 L	Philips Lighting Electronics (Xiamen) Co., Ltd. (Xiamen, China)
Three-dimensional printer	FLASH DK2	Guangzhou flash Information Technology Co., Ltd. (Guangzhou, China)
Regulated power supply	MS-305d	Dongguan Maihao Electronic Technology Co., Ltd. (Dongguan, China)
Voltage resistance equipment meter	Keithley 2400	Teck Technology (China) Co., Ltd. (Shanghai, China)
Resistance metering equipment	TH2829C	Changzhou Tonghui Electronic Co., Ltd. (Changzhou, China)
Scanning electron microscope	FESEM S4800	Hitachi Company (Tokyo, Japan)

**Table 3 sensors-22-03425-t003:** Comparative total strain output and sensitivity of M3 and L4 regarding the simulations and experimental test, respectively.

Model	Simulations					Experiment
	Strain on the Backing Surface ε		Strain on the Adhesive Surface με		Total Strain με	Sensitivity mV/V
	Path01	Path02	Path03	Path04		
M3	11.6	11.4	−10.6	−10.4	44.00	0.101
L4	50.1	49.4	30.5	29.8	39.20	0.0856
Change					12.2%	17.99%

## Data Availability

Not applicable.

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
