# Peer review of "High-Sensitivity Pressure Sensors Based on a Low Elastic Modulus Adhesive"

_sensors, 2022, doi:10.3390/s22093425_

Round 1

Reviewer 1 Report

In the manuscript “High Sensitivity Pressure Sensors Based on Low Elastic Modulus Adhesive” the authors report on the realization of pressure sensors with double-sided sensitive grids. By means experimental methods and finite element simulations, the authors compare the sensitivity coefficients of pressure-sensitive elements bonded on the glass screen using adhesives of different elastic moduli.  In particular, the authors demonstrated that the sensor with low elastic modulus adhesive shows an higher sensitivity. In my opinion, this manuscript presents some interesting results, but I have some points which I think deserve to be better explained or modified before the publication of the manuscript:

  • In the section 2 – testing process, the authors write “ that the glass surface not pasted with the sensor faced upward”. Please clarify better this assertion.
  • In the section 3 “Result and discussion” the authors assert that the surface of film is smooth without obvious agglomeration. Can the author provide additional information about the order of magnitude of roughness?
  • In the manuscript it is not very clear which are the four resistance of Wheatstone full bridge and why the authors prefer to use this configuration. Also, about this aspect, the first paragraph of the supplementary is not very clear and punctuation is missing. I suggest the authors to review it.
  • At pag 6 the authors write “the mean full-bridge voltage output of the sensor (M3) was found to be 0.000303 V when attached to a 3M double-sided tape, and resulting in a sensitivity of 0.101 mV/V, as deduced from the Figure S2”. How do the authors deduce the sensitivity from the figure S2? It would be appropriate to clarify this aspect in the manuscript.
  • how are the sensitivity values obtained in these experiments compared to those reported in the literature?
  • At pag 7 the authors write “When the 3M double-sided adhesive was applied to the sensor, the sensitive grids on the non-adhesive side of the sensor sensed tensile strain, and the other sides of the sensor sensed compressive strain. Whereas, the sensitive grid on the both side of the sensor sensed tensile strain when Loctite 401 adhesive applied to the sensor”. Why is there a difference between the two adhesives? Why with Loctite 401 is there only a tensile strain on both side?
  • Finally, all the paper need an accurate revision

Reviewer 2 Report

The manuscript entitled "high sensitivity pressure sensors based on low elastic modulus adhesive" by Xu et al. presents the sensitivity coefficients of a pressure sensor with double-sided sensitive grids adhered by the high and low elastic modulus adhesive. The manuscript is written based on some experimental and simulation (finite element) results. According to the title of manuscript, it is expected to see some important parameters of pressure sensor such as quality factor, detection limit, dynamic range, so on, whereas this focuses on the sensitivity. Therefore, a key novelty and thorough investigations are needed, the overall analysis is not enough and the quality of the manuscript as it stands, is not sufficient for publication. The results should be compared with previous reports. The quality of some figures should be enhanced. 

Reviewer 3 Report

In this paper, adhesives with different elastic moduli were used to bond pressure-sensitive elements of double-sided sensitive grids. The authors found that the low elastic modulus adhesive produced tensile and compressive strains on both sides of the sensitive grids. This paper contains interesting points, but some explanations are unclear and insufficient. The authors are recommended to revise the following points.

  1. In the introduction section, please describe previous reports on pressure sensors similar to those of this study in more detail. Could the authors specify the issues of the previously reported sensors?

  1. In Fig.3(b), please add the tiles and units of the vertical and horizontal axes and add the scale of the vertical axis.

Round 2

Reviewer 2 Report

The authors have answered partially my requests. 

The manuscript is written based on some experimental and simulation (finite element) results. According to the title of manuscript, it is expected to see some important parameters of pressure sensor such as quality factor, detection limit, dynamic range, so on, whereas this focuses on the sensitivity.

A key novelty and thorough investigations are needed, the overall analysis is not enough and the quality of the manuscript as it stands, is not sufficient for publication.

The results should be compared with previous reports.

I give the authors another opportunity to correct those issues. 
